# Molecular Cloning and Functional Characterization of the 5′ Regulatory Region of the *SLC11A1* Gene from Yaks

**DOI:** 10.3390/ani13233615

**Published:** 2023-11-22

**Authors:** Yuqing Chong, Liping Wang, Bo Wang, Zhendong Gao, Ying Lu, Weidong Deng, Dongmei Xi

**Affiliations:** 1Faculty of Animal Science and Technology, Yunnan Agricultural University, Kunming 650201, China; 2022004@ynau.edu.cn (Y.C.); 13888796612@163.com (B.W.); zander_gao@163.com (Z.G.); yinglu_1998@163.com (Y.L.); dengwd@ynau.edu.cn (W.D.); 2Faculty of Biological Engineering, Xinxiang University, Xinxiang 453003, China; wangliping52071038@xxu.edu.cn

**Keywords:** Zhongdian yak, *SLC11A1* gene, 5′ regulatory region, luciferase reporter gene, transient transfection

## Abstract

**Simple Summary:**

Due to a lack of understanding of the 5′ regulatory region of the *SLC11A1* gene in the yak population, we conducted a comprehensive analysis of the regulatory region of the yak *SLC11A1* gene by cloning gene sequences of different lengths. Through transient transfection experiments, we confirmed the promoter activity of four distinct-length sequences within the 5′ regulatory region of the yak *SLC11A1* gene in the RAW264.7 cell line. Moreover, our analysis identified six novel SNPs in the 5′ regulatory region sequence of the *SLC11A1* gene in yaks. Bioinformatic analysis emphasized the proximity of the −1057 G>T and −127 G>A loci to the transcription factor binding sites NF-1 and NF-1/L. This observation suggests their potential impact on NF-1 and NF-1/L transcription factor binding. In summary, this study serves as a foundational reference for future research on the connection between promoter region polymorphism of the yak *SLC11A1* gene and disease resistance. It also lays the foundation for investigating the transcriptional regulation mechanisms of this gene, and opens the door to further exploration in this important field.

**Abstract:**

The solute transport protein family 11 A1 (*SLC11A1*), also recognized as natural resistance-associated macrophage protein 1 (*NRAMP1*), represents a transmembrane protein encoded by the *SLC11A1* gene. A variety of prior investigations have illuminated its involvement in conferring resistance or susceptibility to bacterial agents, positioning it as a promising candidate gene for breeding disease-resistant animals. Yaks (*Bos grunniens*), renowned inhabitants of the Qinghai-Tibet Plateau in China, stand as robust ruminants distinguished by their adaptability and formidable disease resistance. Notwithstanding these unique traits, there is scant literature on the *SLC11A1* gene in the yak population. Our inquiry commences with the cloning of the 5′ regulatory region sequence of the Zhongdian yak *SLC11A1* gene. We employ bioinformatics tools to identify transcription factor binding sites, delineating pivotal elements like enhancers and cis-acting elements. To ascertain the promoter activity of this region, we amplify four distinct promoter fragments within the 5′ regulatory region of the yak *SLC11A1* gene. Subsequently, we design a luciferase reporter gene vector containing four site-specific deletion mutations and perform transient transfection experiments. Through these experiments, we measure and compare the activity of disparate gene fragments located within the 5′ regulatory region, revealing regions bearing promoter functionality and discerning key regulatory elements. Our findings validate the promoter functionality of the 5′ regulatory region, offering preliminary insights into the core and principal regulatory segments of this promoter. Notably, we identified single nucleotide polymorphisms (SNPs) that may be associated with important regulatory elements such as NF-1 and NF-1/L. This study provides a theoretical framework for in-depth research on the function and expression regulation mechanism of the yak *SLC11A1* gene.

## 1. Introduction

The bovine *SLC11A1* gene has emerged as a focal point of research, particularly as a pivotal candidate gene in investigations pertaining to disease resistance and susceptibility, including ailments such as bovine brucellosis and mastitis [1,2,3,4]. An inaugural milestone in this domain was marked by Joo et al. [5], who reported a noteworthy observation—namely, that mastitis-resistant cattle exhibited heightened mRNA expression levels of the *SLC11A1* gene in comparison to their susceptible counterparts. A previous study identified single nucleotide polymorphisms (SNPs) in the bovine *SLC11A1* gene that are associated with susceptibility to tuberculosis (TB) [6], while in a Chinese Holstein–Friesian cattle population, associations with TB susceptibility/resistance were found with SNPs in exon 11, intron 5, and intron 9 [7]. Additionally, polymorphisms in the 3′ UTR of the bovine *SLC11A1* gene, which have been shown to cause changes to the lengths of two microsatellites (MS1 and MS2), have been identified in different breeds of cattle [8,9,10]. Interestingly, an increase in the length of the MS1 microsatellite has been described to reduce traits indicative of TB infection in African zebu cattle [11]. Despite such significant revelations, Guo et al. studied the polymorphism of exon 10, intron 9, and intron 11 of the *SLC11A1* gene in 771 Chinese Holstein cattle, and found four unlinked SNPs, namely 6067 (A/G) and 6358 (C/T) in intron 9, 7155 (A/G) in exon 10, and 7809 (A/T) in intron 11. The correlation analysis of the different genotypes and the incidence rate of mastitis showed that 6358 (C/T) and 7155 (A/G) of the *SLC11A1* gene had significant effects on the number of somatic cells and milk production (*p* < 0.05). There are a total of 16 random combinations of haplotypes in this population, and the CAAA haplotype combination has been preliminarily determined to be an advantageous haplotype. Individuals with this haplotype have low somatic cell numbers and high milk production, and can be used as candidate gene markers in disease resistance breeding screening for cow mastitis [12]. Nam et al. (2010) used quantitative PCR (RT-PCR) to investigate the differential expression level of *SLC11A1* gene mRNA in peripheral blood mononuclear cells of mastitis-resistant and -susceptible cow herds. For the first time, they reported that the mRNA expression of the *SLC11A1* gene in mastitis-resistant herds was higher than that in susceptible herds, and cows with high mastitis resistance can be screened based on this difference [13]. Bagheri et al. (2016) studied the relationship between the *SLC11A1* gene structure and the genotype in 135 Holstein cows, and found that there was a site mutation in exon 11 of the gene. The three genotypes formed by this point mutation were significantly related to the incidence rate of mastitis in Holstein cows, and this result shows that *SLC11A1* is a candidate gene for mastitis resistance [14]. Gao et al. (2017) applied gene targeting editing technology to insert the *SLC11A1* gene into fetal fibroblasts of cattle. Through somatic transfer, transgenic cattle with strong resistance to tuberculosis were ultimately obtained, further indicating that the *SLC11A1* gene is an important candidate gene in disease resistance breeding research [15]. 

The 5′ regulatory region of a gene plays a pivotal role in directly or indirectly regulating gene transcription. In addition to serving as the docking site for RNA polymerase, this regulatory region encompasses numerous transcription factor binding sites. Promoters serve as the controllers of transcription initiation and dictate the magnitude of gene expression. Earlier investigations have revealed that the promoter region of the bovine *SLC11A1* gene deviates from the conventional presence of TATA boxes or CAAT boxes [16]. This deviation represents a phenomenon of conserved element loss during transcription initiation, which is a common feature with the corresponding promoter region of the human SLC11A1 gene [17]. However, specific tissue-specific transcription factor binding motifs can be identified within the promoter region of the *SLC11A1* gene in both humans and cattle, enabling transcription initiation even in the absence of TATA and CAAT boxes.

Recent research on the promoter region of the human *SLC11A1* gene has unveiled (GT)n microsatellite polymorphic sites, suggesting their potential to influence DNA conformation and, consequently, gene transcription and downstream expression [18]. Studies on the mouse *SLC11A1* gene promoter have revealed the presence of more than two transcription initiation start elements, which counteract the deletion of the TATA box. The expression of this gene is characterized by a high degree of cell specificity, with primary expression occurring in macrophages [19]. Holly et al. explored the promoter activity of the mouse *SLC11A1* gene and transfected it into RAW264.7 cells [20]. Their findings highlighted significant promoter activity, particularly in the −17 bp to +34 bp region of the core promoter. They also identified influential factors that affect gene promoter activity, including the SPL binding site within the core promoter region, Miz-1, and GC box. These studies have laid the foundation for further investigations into the expression and regulatory mechanisms of the *SLC11A1* gene. The bovine *SLC11A1* gene stands as a pivotal player in regulating the susceptibility of cattle to infectious diseases. However, research in this domain remains somewhat limited, particularly concerning the promoter of the bovine *SLC11A1* gene. A special report [21] describes the cloning of a 2138 bp long bovine *SLC11A1* promoter gene sequence, which was subsequently recombinant and transfected into the RAW264.7 cell line. The results of this study indicated relatively weak promoter activity. Regrettably, research on the promoter of the yak *SLC11A1* gene has remained conspicuously absent from the scientific literature.

Promoters, conventionally situated upstream of genes, hold a crucial role in identifying RNA polymerase and forming binding interactions with it, thereby promoting gene transcription processes. These promoters also encompass an array of binding sites typically located upstream of the 5′ terminus of the gene, which, in turn, control the participation of protein factors. Promoters are typically limited to a range of approximately 35 base pairs from the transcription initiation site, exerting the author’s control over gene transcription efficiency. The selection of appropriate promoters stands paramount, particularly in augmenting the expression of foreign genes [22]. Promoters, in essence, dictate the sequence of transcription initiation, thereby orchestrating the temporal and quantitative aspects of gene expression, a facet equally germane to prokaryotes and eukaryotes alike. The binding strength between the promoter and RNA polymerase directly affects the frequency of initiating events. Indeed, the crux of gene expression regulation through transcription hinges upon the dynamic interplay between the promoter, RNA polymerase, and other trans-acting factors. It is imperative to recognize that promoters themselves carry distinct signals integral to life activities, as their DNA sequences markedly differ from those of the transcriptional regions. The pivotal role played by promoters in gene transcription and subsequent expression underscores their significance in molecular biology. As Zhang et al. (2009) pointed out [22], comprehending the structure of promoters, devising efficient methods for their identification within the genome, and deciphering the wealth of information they contain represent invaluable research pursuits. With advancements in computer technology, computational simulations have become invaluable tools for predicting and analyzing biological information, offering rapid and cost-effective solutions [23]. Such predictive information often guides researchers in refining their scientific inquiries, enhancing research efficiency, and providing a roadmap for subsequent validation studies.

## 2. Materials and Methods

### 2.1. Sample Collection and DNA Extraction

A total of 89 spleen samples were collected from Zhongdian yaks at a slaughterhouse located in Shangri La, within the Diqing Prefecture of Yunnan Province, China. Subsequently, these samples were immediately stored in a refrigerator at −80 °C to preserve them for subsequent experimental research. DNA was extracted from the spleen tissue of Zhongdian yaks using a tissue genomic DNA purification kit provided by Beijing TransGen Biotech Co., Ltd. (Beijing, China).

### 2.2. Cloning and Vector Construction of Yak SLC11A1 Gene 5′ Regulatory Region Sequences of Various Lengths

In order to distinguish the various gene fragment sequences within the 5′ regulatory region of the yak *SLC11A1* gene, we assigned different names to the inserted fragments based on their positions in the gene. These names are as follows: *SLC11A1*-5′ (−1418~+342), *SLC11A1*-5′-1 (−1418~+58), *SLC11A1*-5′-2 (−797~+58), *SLC11A1*-5′-3 (−278~+58), and *SLC11A1*-5′-4 (−205~+58).

Primer design was conducted using these gene sequences, namely *SLC11A1*-5′, *SLC11A1*-5′-1, *SLC11A1*-5′-2, *SLC11A1*-5′-3, and *SLC11A1*-5′-4, each featuring different lengths within the 5′ regulatory region. Additionally, the vector sequences from PGL3 were integrated into the primer design. These primers were synthesized by Shanghai Sangon Biotech Co., Ltd (Shanghai, China), and constructed to incorporate two critical restriction sites, namely *KpnI* and *XhoI*. The expected lengths of the amplified fragments were as follows: 1760 bp, 1476 bp, 855 bp, 336 bp, and 263 bp (Table 1). For PCR reactions, the following conditions were meticulously followed: initial denaturation at 95 °C for 5 min, denaturation at 95 °C for 30 s, annealing at 59 °C for 30 s, extension at 72 °C for 1 min and 30 s, final extension at 72 °C for 5 min. This process was repeated for a total of 35 cycles. Subsequently, the gel purification kit from Qingke Biological Co., Ltd (Beijing, China). was employed for efficient gel purification and recovery of the amplified fragments. Finally, the PCR amplification products were sent to Guangzhou Vipotion Biotechnology Co., Ltd. (Guangzhou, China) for Sanger sequencing.

### 2.3. BLAST Analysis

The BLAST analysis of sequencing results confirmed that the *SLC11A1*-5′ sequence was successfully amplified, signifying its readiness for subsequent sequence alignment analysis. Simultaneously, this analysis validated the successful cloning of the promoter gene sequences of *SLC11A1*-5′-1, *SLC11A1*-5′-2, *SLC11A1*-5′-3, and *SLC11A1*-5′-4 into the PGL3 vector. These vectors were named PGL3/*SLC11A1*-5′-1, PGL3/*SLC11A1*-5′-2, PGL3/*SLC11A1*-5′-3, and PGL3/*SLC11A1*-5′-4. These constructs were poised for utilization in subsequent experiments involving cell transient transfection.

### 2.4. Investigating the 5′ Regulatory Region Sequence of the Yak SLC11A1 Gene through Bioinformatics Analysis

Similarity analysis was conducted employing the BLAST program within the NCBI database, aligning the 5′ regulatory region sequence of the yak *SLC11A1* gene against its counterparts in the *SLC11A1* genes of diverse species, including *Bos taurus* (cattle) (Gene ID: 282470), *Bos indicus* (zebu cattle) (109570553), bison (105003764), *Bubalus bubalis* (water buffalo) (102405015), *Ovis aries* (sheep) (443365), and *Canis lupus familiaris* (dog) (478909). 

The predictive TATA box was identified via the online tool available at http://bioinfo.itb.cnr.it/~Webgene/wwwHC_tata.html (accessed on 10 October 2023). Foreseeing CpG Islands was accomplished utilizing the online analysis software accessible at http://www.ebi.ac.uk/Tools/seqstats/emboss_cpgplot (accessed on 13 October 2022). Further investigations involved the utilization of the amplified promoter sequence from the yak *SLC11A1* gene, which served as a reference. Online tools available at http://www.fruitfly.org/seq_tools/promoter.html (accessed on 19 October 2022) and http://bioinfo.itb.cnr.it/~Webgene/wwwHC_polya.html (accessed on 20 October 2022) were then deployed to prognosticate promoter regions, ascertain transcription start sites, delineate termination sites, and identify transcription factor binding sites. In parallel, predictions regarding duplicate components within the sequence were made, utilizing the RepeatMask program accessible at http://www.repeatmasker.org/cgi-bin/WEBRepeatMasker (accessed on 30 October 2022) and https://www.itb.cnr.it/sun/webgene/ (accessed on 1 November 2022).

### 2.5. Assessment of Single Nucleotide Polymorphisms (SNPs) Variability within the 5′ Regulatory Region of the Yak SLC11A1 Gene

The amplified products of the 5′ regulatory region of the *SLC11A1* gene in yaks were submitted to the Guangzhou Vipotion Biotechnology Co., Ltd. for DNA bidirectional sequencing. Subsequently, the SeqMan program within the DNAStar software package (version 7.1) was employed to compare and analyze the sequencing results, with a focus on the identification of SNP mutations.

Utilizing online tools such as MatInspector professional 7.2.2, the sequence of the 5′ regulatory region of the yak *SLC11A1* gene was subjected to comprehensive analysis. For any novel SNPs uncovered, a preliminary assessment of their biological significance was conducted based on the outcomes of sequence prediction analyses. Particular attention was given to assessing the precise location of these SNPs, determining whether they reside within or near crucial expression regulatory sites or within repetitive sequences. SNPs deemed to play a pivotal role in the regulatory landscape were singled out for further investigation.

### 2.6. Cell Culture, Transfection, and Dual Luciferase Reporter Gene Assay

In a sterile environment, the cell culture medium was removed from the culture flask and one to two gentle washes with PBS were performed. Subsequently, the cell flask’s base was rinsed with trypsin solution, followed by the removal of trypsin solution. The flask was then placed within a 37 °C incubator for a duration of 2–3 min. Subsequently, an appropriate volume of fresh culture medium enriched with serum was introduced to terminate the trypsin action. To ensure uniform mixing, a pipette was utilized to disintegrate cell clumps, followed by the transfer of the cell suspension into a new culture flask in accordance with the designated dilution ratio. The cells were then incubated within a 37 °C constant-temperature incubator under 5% CO_2_-saturated humidity.

The experimental grouping for recombinant and empty vectors featuring various-length fragments of the 5′ regulatory region of the *SLC11A1* gene was as follows: (1) empty vector group, (2) PGL3/*SLC11A1*-5′-1 (1476 bp), (3) PGL3/*SLC11A1*-5′-2 (855 bp), (4) PGL3/*SLC11A1*-5′-3 (336 bp), (5) PGL3/*SLC11A1*-5′-4 (263 bp), (6) PGL3/*SLC11A1*-5′-1 (1476 bp) + LPS/IFN-γ, (7) PGL3/*SLC11A1*-5′-2 (855 bp) + LPS/IFN-γ, (8) PGL3/*SLC11A1*-5′-3 (336 bp) + LPS/IFN-γ, and (9) PGL3/*SLC11A1*-5′-4 (263 bp) + LPS/IFN-γ. The experimental procedure was as follows: (1) Cells from the above-mentioned groups were harvested to preparate single-cell suspensions. (2) The number of cells was counted, and the cell density was adjusted to 8 × 10^5^ cells/mL. (3) A 24-well plate was inoculated with 100 µL of the cell suspension (density: 8 × 10^5^ cells/mL) and cultured at 37 °C with 5% CO_2_. (4) In a 125 µL DMEM (Dulbecco’s modified Eagle medium) serum-free medium, 0.8 µg of recombinant plasmid and 0.8 µg of pGL3 basic plasmid were added (a serum-free culture medium containing 5 μg/mL LPS (lipopolysaccharid) and 10 ng/mL IFN-γ in DMEM was used in groups 6, 7, 8, and 9). (5) In a 125 µL DMEM serum-free medium, 2 µL of Lipofectamine™ 2000 (Beijing, China) was added (a mixture containing 5 μg/mL LPS and 10 ng/mL IFN-γ (interferon-γ) in DMEM serum-free medium was used in groups 6, 7, 8, and 9). (6) Following steps 5 and 6, the plasmid was diluted and Lipofectamine™ 2000 was added, and left at room temperature for 20 min to form a complex (plasmid/Lipofectamine™ 2000). (7) Next, 250 µL of the complex (plasmid/Lipofectamine™ 2000) was added to the well containing cells and the culture medium. (8) The complex was incubated for 5 h, then fresh complete culture medium was introduced for transfection. (9) The plate was washed twice with PBS, and 100 µL of passive lysis buffer (PLB) was added to each cell well, and cell lysate was collected. (10) For each sample, 20 µL of cell lysate was added to the luminescence plate and the background value was recorded for 2 s using a bioluminescence detector (GloMax) (Beijing, China). (11) Then, 100 µL of LAR II (Luciferase Assay Reagent II) solution was processed into each sample, and recorded for 2 s. (12) After recording, 100 µL of Stop and Glo^®^ Reagent was introduced into each sample, and placed within a luminescence detector for recording.

### 2.7. Statistical Analysis

Utilizing SPSS 22.0 software, we computed both the mean and standard deviation of the experimental data within each group. Subsequently, one-way ANOVA was employed to scrutinize the data for any discernible significant distinctions across the experimental groups. The presentation of our findings was accomplished using GraphPad Prism 5. For sequence alignment, the Clusterx program (version 1.83) was enlisted to ensure precision in our analysis. The determination of genotype, allele, and haplotype frequencies was executed via the utilization of the online software (http://analysis.bio-x.cn/myAnalysis.php) (accessed on 9 March 2023). To delve into the intricate landscape of polymorphisms within our dataset, we harnessed a specialized program designed for the calculation of polymorphism information content (PIC) values. Moreover, the fundamental assessment of the Hardy–Weinberg equilibrium (H–W) was conducted employing the PopGen32 software.

## 3. Results 

### 3.1. Variation of Gene Sequences with Different Lengths in the 5′ Regulatory Region of Yak SLC11A1 Gene 

The electrophoresis results of PCR products are illustrated in Figure 1, revealing distinct and well-defined bands corresponding to gene sequences with different lengths in the 5′ regulatory region of the yak *SLC11A1* gene. These sequences include *SLC11A1*-5′ (1760 bp), *SLC11A1*-5′-1 (1476 bp), *SLC11A1*-5′-2 (855 bp), *SLC11A1*-5′-3 (336 bp), and *SLC11A1*-5′-4 (263 bp).

### 3.2. Enzymatic Digestion Profiling for Verification of Recombinant Plasmids

Four recombinant plasmids, PGL3/*SLC11A1*-5′-1, PGL3/*SLC11A1*-5′-2, PGL3/*SLC11A1*-5′-3, and PGL3/*SLC11A1*-5′-4, were identified using restriction endonucleases *KpnI* and *XhoI*, respectively. The electrophoresis results showed consistency with the expected fragments, as shown in Figure 2.

### 3.3. Sequencing Identification

The results of BLAST analysis showed that the sequence similarity of our target fragment (*SLC11A1*-5′ PCR products and the four recombinant plasmids, namely PGL3/*SLC11A1*-5′-1, PGL3/*SLC11A1*-5′-2, PGL3/*SLC11A1*-5′-3, and PGL3/*SLC11A1*-5′-4) to the corresponding fragment of the published *SLC11A1* gene sequence surpasses the threshold of 99%. This result affirmed the successful cloning of the yak *SLC11A1* gene 5′ regulatory region sequence in this endeavor and the construction of recombinant plasmids PGL3/*SLC11A1*-5′-1 and PGL3/*SLC11A1*-5′-2. These plasmids were instrumental for subsequent bioinformatics analyses of the 5′ regulatory region sequence of the bovine *SLC11A1* gene and for facilitating the expression of the promoter EGFP reporter gene in PGL3/*SLC11A1*-5′-3 and PGL3/*SLC11A1*-5′-4.

### 3.4. Bioinformatics Exploration of the 5′ Regulatory Region Sequence of the Yak SLC11A1 Gene

We conducted a BLAST analysis using the cloned promoter sequence of the yak *SLC11A1* gene and sequences from other species in the GenBank database, revealing a noteworthy degree of sequence homology with various species. Intriguingly, those regions exhibiting relatively conserved homology were predominantly situated within the proximal promoter region. In contrast, those regions with marked sequence disparities were frequently located in the distal promoter region. The following homology percentages were observed: 99% with cattle, 99% with zebu, 98% with bison, 98% with water buffalo, 90% with sheep, and 75% with dogs.

In the 5′ regulatory region of the yak *SLC11A1* gene, we did not identify a typical TATA box. Instead, atypical TATA boxes were identified, distributed proximally near the transcription start site (−148 bp~−671 bp), as well as within other regulatory regions, such as TCTACCCTG (−1255 bp~−1246 bp), GGGATAAGG (−947 bp~−938 bp), TTTATTACT (−671 bp~−662 bp), CCTATTACTT (−428 bp~−419 bp), ATTACTTCCC (−425 bp~−416 bp), and TGAATCTTCA (−157 bp~−148 bp). Moreover, the Methprimer prediction results showed that there were no CpG islands in the region.

Our analysis pinpointed a promoter region within this region, consisting of 50 bases (GACCCAAGATTAAAGGGAGAGACCTGACTGCTTACAGGGTGAGGG), with the bold red letter ‘A’ signifying the transcription start site (Figure 3). Subsequently, we conducted a comprehensive examination of this region and identified many transcriptions factor binding sites. Notable binding sites included three alpha-CBF sites (−1199 bp, −911 bp, −828 bp), three alpha-IRP sites (−1199 bp, −911 bp, −828 bp), three CBF-B sites (−1199 bp, −911 bp, −828 bp), two alpha-CP1 sites (−1199 bp, −911 bp), two AP-1 sites (−1273 bp, −1150 bp), two NF-Y sites (−911 bp, −828 bp), and one AP-2 site (−173 bp), in addition to a single c-Myc site (−713 bp). Furthermore, multiple GR binding sites were identified at positions such as −1286 bp, −1084 bp, −997 bp, −701 bp, −676 bp, −363 bp, −179 bp, and +161 bp. Similarly, several SP1 binding sites were observed at positions like −131 bp, −1193 bp, −690 bp, −492 bp, +19 bp, +148 bp, and +269 bp. We also detected multiple NF-1 sites at locations like −1391 bp, −1231 bp, −1062 bp, −92 bp, −123 bp, and −126 bp, as well as NF-1/L sites at positions like −1024 bp, −93 bp, −926 bp, −212 bp, −133 bp, and −54 bp. Additionally, multiple NF-E binding sites were identified at locations like −1401 bp, −1283 bp, −1199 bp, −1001 bp, −347 bp, and −76 bp.

Intriguingly, these binding sites were conserved at corresponding positions in the gene sequences of cattle, zebu, bison, water buffalo, sheep, and dogs, indicating a common regulatory framework among these species (Figure 4). 

The prediction results of the repetitive elements revealed the existence of three short and dispersed duplicate elements: BTALΜL1 (−642 bp~−450 bp), BOVTA (−625 bp~−443 bp), and BTCS (−565 bp~−432 bp). In addition, we also identified the duplicate DNA element, namely Charlie 8/hAT Charlie (−1154 bp~−1077 bp).

### 3.5. Analysis of Single Nucleotide Polymorphism Variation Sites in the 5′ Regulatory Region of the Yak SLC11A1 Gene

Utilizing bioinformatics analysis software and sequence scrutiny from bidirectional sequencing data, we successfully identified and characterized a total of six SNPs within the 5′ regulatory region of the yak *SLC11A1* gene. These SNPs are accurately located at positions −1264 A>C, −1237 A>G, −1057 G>T, −127 G>A, −96 A>G, and −50 G>A (Figure 5). 

Our statistical analysis, encompassing gene and genotype frequency polymorphism information content, indicated that the homozygous genotypes dominated at these six loci. Moreover, the genotype frequencies for each locus demonstrated a significant low mutation rate, with rare genotypes occurring at frequencies as low as 0.045 and 0.056—in other words, these loci were almost fixed alleles. This indicates a relatively conservative and limited polymorphism within the 5′ regulatory region of the *SLC11A1* gene in yaks. Notably, the bioinformatics analysis emphasized that the −1057 bp G>T and −127 bp G>A loci were situated in close proximity to transcription factor binding sites, specifically NF-1 and NF-1/L. Except for the −1264 bp A>C and −96 bp A>G loci with moderate polymorphism, the remaining four loci exhibited lower polymorphism, emphasizing the overall conservatism of this gene region. It is worth noting that the Hardy–Weinberg equilibrium test revealed that, except for the −1264 bp A>C and −50 bp G>A sites, which exhibited *p* values ≥ 0.05 and thus adhered to the Hardy–Weinberg equilibrium, the remaining four sites did not conform to the Hardy–Weinberg equilibrium (Table 2).

Further genetic analysis of these six SNPs resulted in the identification of a total of 16 different haplotypes. Among these, the haplotype ‘AAGGAG’ emerged as the dominant one, with the highest frequency of 0.854 (Table 3). 

Additional examination of the linkage disequilibrium for these six SNPs indicated that half of the loci (−1264 A>C vs. −1237 A>G, −1264 A>C vs. −1057 G>T, −1264 A>C vs. 127 G>A, −1237 A>G vs. −127 G>A, −1237 A>G vs. −50 G>A, −1057 G>T vs. −127 G>A, −1057 G>T vs. −50 G>A, and −127 G>A vs. −96 A>G) were in linkage equilibrium, while the remaining half exhibited linkage disequilibrium (Table 4).

The results obtained from the RepeatMask program analysis showed that none of the six SNPs were consistent with the repeat sequence. Notably, the −1057 bp G>T locus was positioned a mere five bases away from the transcription factor binding site NF-1 (−1062), while the −127 bp G>A mutation was at a distance of six bases from NF-1/L (−133).

### 3.6. Cell Transfection and Fluorescence Detection Outcomes for Varied Segments within the 5′ Regulatory Region Sequence of the Yak SLC11A1 Gene

The results of cell transfection experiments demonstrated the expression of the recombinant plasmids PGL3/*SLC11A1*-5′-1, PGL3/*SLC11A1*-5′-2, PGL3/*SLC11A1*-5′-3, and PGL3/*SLC11A1*-5′-4 in RAW264.7 cells post-transfection. This observation indicated the promoter functionality of different-length gene sequences within the regulatory region of the yak *SLC11A1* gene in the RAW264.7 cell line. However, changes in fluorescence intensities suggested differing relative fluorescence activities among these gene sequences. Using sea kidney luciferase as an internal control, the activity of each promoter was standardized (Figure 6).

The results showed that the reporter genes constructed with different amplified 5′ regulatory region lengths exhibited promoter activity significantly higher than the empty vector (*p* < 0.05). Furthermore, the activity of the reporter gene containing a 1476 bp vector surpasses that containing a 263 bp vector (*p* < 0.05). This phenomenon highlights the variability of promoter activity detected in different missing fragments, indicating the presence of key regulatory elements in specific regions. Upon induction with 5 μg/mL LPS/10ng/mL IFN-γ, the fluorescence activity of the recombinant vector containing luciferase was significantly increased (*p* < 0.05). These results emphasized the basic promoter activity of the -205~58 gene sequence fragments.

## 4. Discussion

A detailed sequence analysis of the 5′ regulatory region of the *SLC11A1* gene revealed the presence of six atypical TATA boxes, which were located upstream of the transcription initiation site and lacked the typical TATA box motifs. Employing online software analysis, we aligned the promoter region sequence of the yak *SLC11A1* gene with those of cattle, zebu, bison, water buffalo, sheep, and dogs. The results revealed that sequence fragments with relatively conservative homology were predominantly found in the proximal promoter region. Conversely, fragments exhibiting significant divergence were typically located in the distal promoter region. Furthermore, some binding sites were found to be conserved across gene sequences of the different species mentioned above. These findings suggested that these transcription factors may play a key regulatory role in the transcription and expression levels of the yak *SLC11A1* gene.

Interestingly, the distal promoter region of the *SLC11A1* gene in yaks contains three scattered repetitive elements, BTALΜL1 (−642 bp~−450 bp), BOVTA (−625~−443 bp), and BTCS (−565~−432 bp), in addition to duplicate DNA elements (Charlie 8/hAT Charlie, −1154 bp~−1077 bp). It is worth noting that the presence of repetitive elements in gene sequences may be due to the insertion of reverse transcripts. Such repetitive elements can serve as a protective mechanism that has evolved over time to safeguard the promoter, exon, and other regulatory elements from gene mutations that could potentially compromise their functionality. However, the specific roles and mechanisms associated with these repetitive elements remain areas of ongoing investigation and exploration.

In order to delve deeper into the molecular complexity of controlling the transcriptional regulation of the yak *SLC11A1* gene, it is imperative to identify the key regulatory elements and other important sequences within its 5′ regulatory region. In this experiment, the EGFP reporter gene was selected as a vehicle to examine the promoter functionality inherent within the 5′ regulatory region of the yak *SLC11A1* gene. The EGFP reporter gene boasts several advantages, including minimal background interference in animal cells, the ability to observe directly under long wavelength ultraviolet or visible light, and ease of detection and high sensitivity. Previous studies have indicated that the target tissues for *SLC11A1* gene expression are primarily the spleen, lungs, and related organs. The target cells include macrophages and reticuloendothelial cells derived from these organs, as well as peripheral white blood cells [26,27]. Consequently, in order to comprehensively explore the biological function of the 5′ regulatory region sequence of the yak SLC11A1 gene, the RAW264.7 mouse macrophage cell line was selected.

Through transient transfection, a dual luciferase reporter gene EGFP system was constructed, followed by a luciferase activity assessment. These findings elucidated the promoter activity of these sequences with different lengths in the RAW264.7 cell line, confirming that plasmids spanning from −205 bp to +58 bp have basic promoter activity. Notably, the plasmid spanning from −1418 bp to +58 bp exhibited heightened promoter activity in RAW264.7 cells compared to the −205 bp to +58 bp plasmid (*p* < 0.05). This enhancement indicated the presence of positive regulatory elements in the differential regions between these plasmids. Such a phenomenon may be attributed to the positive regulatory role played by binding sites like alpha-CBF, alpha-IRP, CBF-B, alpha-CP1, AP-1, NF-Y, c-Myc, and NF-1 present in this region, all bolstering the promoter activity of the yak *SLC11A1* gene. However, the exact factors responsible for this positive regulatory effect, as well as whether other factors are involved, still need to be further explored and validated through more targeted mutation analysis.

To delve into the impact of LPS stimulation on the *SLC11A1* gene, LPS induction experiments were conducted, and revealed a significant increase in luciferase activity (*p* < 0.05). In a previous study [22], the effects of different concentrations of LPS on recombinant plasmids p-1748/+58 transfected into RAW264.7 and 293T cell lines was investigated. The findings demonstrated a significant enhancement in the activity of the bovine *SLC11A1* promoter in RAW264.7 cells after LPS induction, with the highest induction effect observed at an LPS concentration of 1000ng/mL, significantly surpassing the blank control group (*p* < 0.05). This observation aligns with our research results. Notably, LPS induction did not significantly affect the promoter activity in the 293T cell line (*p* > 0.05). Although the promoter activity of each recombinant plasmid increased and decreased after LPS induction compared to those without LPS, the increase was statistically significant (*p* < 0.05), while the decrease was not (*p* > 0.05). In our experiment, we employed a 5 μg/mL concentration of LPS for induction, where a significant difference (*p* < 0.05) compared to the blank control was observed. The promoter activity of each recombinant plasmid induced by LPS exhibited a substantial increase compared to those without LPS (*p* < 0.05).

The transfection of RAW264.3 and 293T cell lines indicated that the region of the bovine *SLC11A1* gene exhibiting basal promoter activity extended from the 5′ regulatory region of -89 bp to +58 bp [22]. Consistent with Pugh and Tjian [28], SP1 emerges as a pivotal player in the regulation of gene expression, actively participating in the regulation of transcription initiation, thereby compensating for the impact of promoter structures devoid of typical TATA boxes on promoter function. Holly et al. reported that the transcriptional regulation of the mouse *SLC11A1* gene depends on the SPL binding site [20]. Similarly, in this study, it was unveiled that within the promoter region of the *SLC11A1* gene in yaks, GR binding sites were located at −371 bp, −194 bp, −150 bp, −134 bp, and 227 bp, NF1 binding sites were located at −228 bp, −169 bp, −142 bp, −56 bp, and 160 bp, alongside transcription factors such as SP1 at −18 bp, −88 bp, 137 bp, and 261 bp, and TGT3 at −20 bp, which were conserved at the corresponding positions in the gene sequences of yellow cattle, zebu, bison, water buffalo, sheep, and dogs. These findings strongly suggest the pivotal role played by SP1 in the basal expression of the yak *SLC11A1* gene.

Comprehensive information on SNPs of the *SLC11A1* gene can be found in various databases, including NCBI and Ensembl. For instance, Bellamy et al. [29] reported the discovery of a (GT)n repeat in the promoter region of humans. However, the bovine *SLC11A1* gene’s SNPs remain less well documented. Martinez et al. [30] did report a C/T mutation at position −20 in the 5′ regulatory region of the *SLC11A1* gene, focusing on Romance cattle and zebu cattle. Nevertheless, these studies, limited to specific cattle herds in specific regions, lack comprehensive representativeness. Therefore, this study employed direct sequencing to analyze the 5′ regulatory region (−1418 bp~+342 bp) of the *SLC11A1* gene in yaks. The polymorphism analysis of the 5′ regulatory region in the yak *SLC11A1* gene unveiled six novel SNP loci. Although these loci may not directly alter protein composition, they may exert regulatory roles in gene transcription and protein synthesis due to the presence of core promoters, distant regulatory elements, and remote regulatory promoter regions located upstream, downstream, proximal, and distal to the transcription starting points. These findings lay the foundation for further exploring the connection between *SLC11A1* gene polymorphism and function of yaks [22]. Sequence analysis using online predictive tools confirmed that these six SNPs loci were not located in repetitive sequences. Notably, −1057 bp G>T was located five bases away from the transcription factor binding site NF-1 (−1062 bp), while −127 bp G>A resides six bases away from NF-1/L (−133 bp). This observation indicated a potential role for this region in gene expression regulation. According to the existing literature, the *SLC11A1* gene is subject to regulation by transcription factors such as SP1, PU.1, NF-1, and NF-1/L [19,20]. Therefore, these loci may play a role in combating pathogenic infections.

## 5. Conclusions

In this study, we conducted a comprehensive analysis of the 5′ regulatory region of the yak *SLC11A1* gene by cloning gene sequences of different lengths, and confirmed the promoter activity of four distinct-length sequences within the 5′ regulatory region of the yak *SLC11A1* gene in the RAW264.7 cell line. Moreover, our analysis identified six novel SNPs in the 5′ regulatory region sequence of the *SLC11A1* gene in yaks. Bioinformatic analysis emphasized the proximity of the −1057 G>T and −127 G>A loci to the transcription factor binding sites NF-1 and NF-1/L. This observation suggested their potential effect on the binding of NF-1 and NF-1/L transcription factors. Collectively, this study serves as a foundational reference for future research concerning the connection between promoter region polymorphism of the *SLC11A1* gene in yaks and disease resistance. 

## Figures and Tables

**Figure 1 animals-13-03615-f001:**
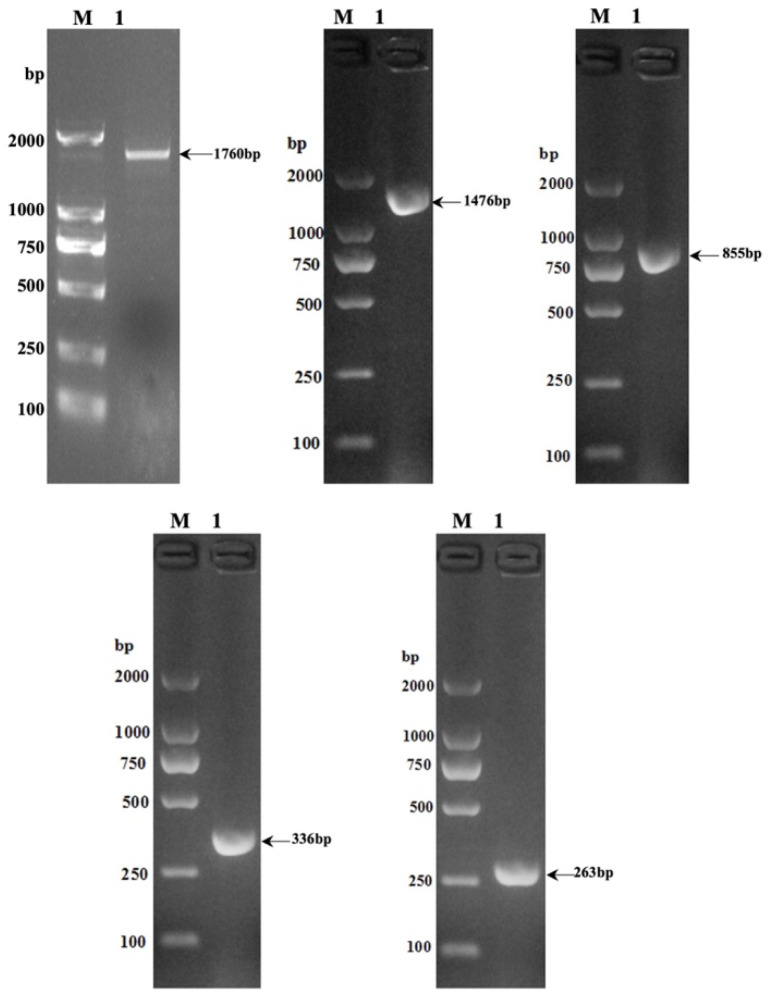
Electrophoresis of PCR products. Starting from the top left: *SLC11A1*-5′, *SLC11A1*-5′-1, *SLC11A1*-5′-2, *SLC11A1*-5′-3, *SLC11A1*-5′-4.

**Figure 2 animals-13-03615-f002:**
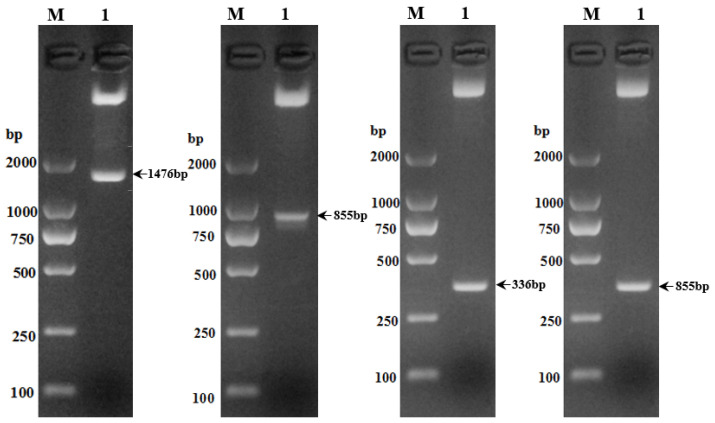
Double electrophoresis of recombinant plasmid with PCR-amplified product electrophoresis. Starting from the left: PGL3/*SLC11A1*-5′-1, PGL3/*SLC11A1*-5′-2, PGL3/*SLC11A1*-5′-3, PGL3/*SLC11A1*-5′-4.

**Figure 3 animals-13-03615-f003:**
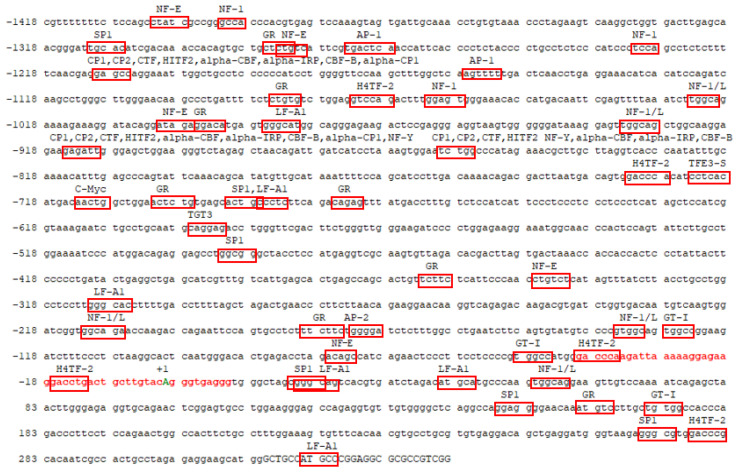
Nucleotide sequence information of the yak *SLC11A1* gene 5′ regulatory region. +1 represents the transcription start site, the red text represents the predicted promoter region, the boxes represent the predicted transcription factor binding sites, and the uppercase letters represent the exon sequences.

**Figure 4 animals-13-03615-f004:**
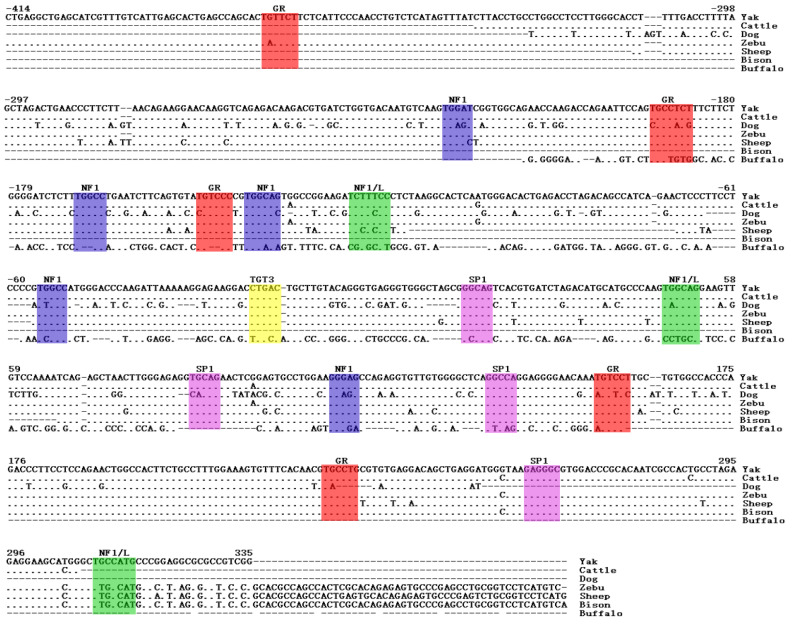
Sequence alignment of *SLC11A1* gene promoter and conservative binding sites of transcription factors in different species. Boxes of different colors are predicted conservative transcription factor binding sites.

**Figure 5 animals-13-03615-f005:**
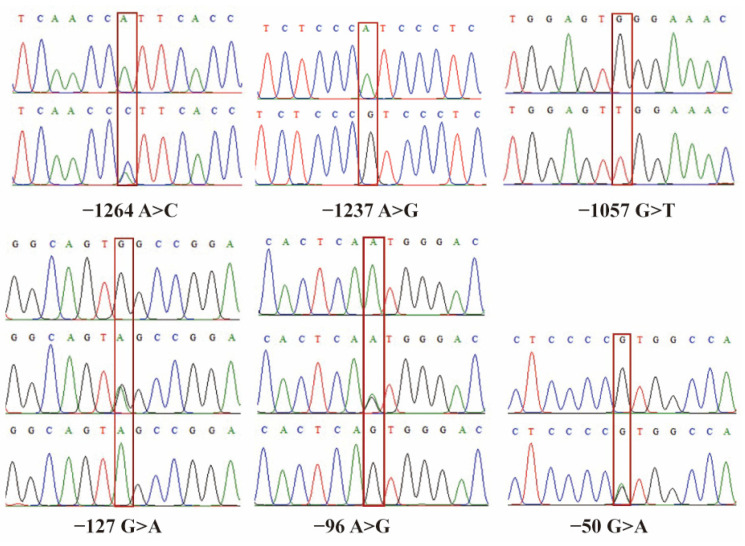
Map of SNPs’ mutation loci in the 5′ regulatory region of the yak *SLC11A1* gene. The rectangular box represents different genotypes, the two peaks on the peak map represent heterozygous genotypes, and the single peak represents homozygous genotypes.

**Figure 6 animals-13-03615-f006:**
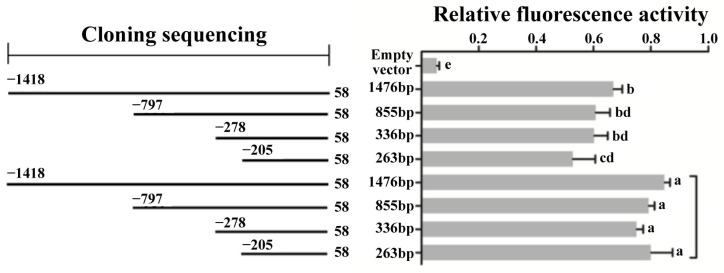
Relative fluorescence activity of the reporter gene with different length fragments after transfection. The difference between experimental groups with completely different letters is significant (*p* < 0.05), while the difference between experimental groups with the same letter is not significant (*p* > 0.05).

**Table 1 animals-13-03615-t001:** Primers used to amplify yak *SLC11A1* gene 5′ regulatory region and 4 mutated gene sequences.

ID	Primers	Sequence (5′-3′)
1	*SLC11A1-5′-F*	GGAAACCATCACAGCCTCCTACC
	*SLC11A1-5′-R*	CCGGGGCACTCTCTGTGCGAGT
2	*SLC11A1-5′-1-KpnI-F*	GGGGTACCGGGTTCCAAGCTTTGGCTC
	*SLC11A1-5′-1-XhoI-R*	CCGCTCGAGGGGCACTCTCTGTGCGAGT
3	*SLC11A1-5′-2-KpnI-F*	GGGGTACCCGTCCATGAGGTCGCAAGT
	*SLC11A1-5′-2-XhoI-R*	CCGCTCGAGGGGCACTCTCTGTGCGAGT
4	*SLC11A1-5′-3-KpnI-F*	GGGGTACCAGACATGCATGCCCAAGTG
	*SLC11A1-5′-3-XhoI-R*	CCGCTCGAGGGGCACTCTCTGTGCGAGT
5	*SLC11A1-5′-4-KpnI-F*	GGGGTACCGCCTGGAAGGGAGCCAGA
	*SLC11A1-5′-4-XhoI-R*	CCGCTCGAGGGGCACTCTCTGTGCGAGT

The underlining indicates the cleavage site.

**Table 2 animals-13-03615-t002:** Genotype of SNPs’ loci in the 5′ regulatory region of the *SLC11A1* gene in yaks.

SNPs	Genotype	Number	Genotypic Frequency	Allele	Number	Allele Frequency	*PIC*	*chi2-HW*	*P-HW*
−1264 A>C	A/A	72	0.809	A	161	0.904	0.261	0.929	0.335
	A/C	17	0.191	C	17	0.096			
−1237 A>G	A/A	85	0.955	A	170	0.955	0.082	101.167	0
	G/G	4	0.045	G	8	0.045			
−1057 G>T	G/G	85	0.955	G	170	0.955	0.082	101.167	0
	T/T	4	0.045	T	8	0.045			
−127 G>A	G/G	80	0.899	G	164	0.921	0.178	45.612	0
	G/A	4	0.045	A	14	0.079			
	A/A	5	0.056						
−96 A>G	A/A	72	0.809	A	156	0.876	0.296	13.477	0
	A/G	12	0.135	G	22	0.124			
	G/G	5	0.056						
−50 G>A	G/G	82	0.921	G	171	0.921	0.135	0.127	0.721
	A/G	7	0.079	A	7	0.79			

The PIC (polymorphism information content) [24,25] greater than 0.5 indicating high polymorphism; 0.5 > *PIC* > 0.25, indicating moderate polymorphism; *PIC* < 0.25, indicating low degree polymorphism. *p* ≥ 0.05, in Hardy–Weinberg equilibrium; *p* < 0.05, in Hardy–Weinberg imbalance.

**Table 3 animals-13-03615-t003:** Haplotype of SNPs’ loci in the 5′ regulatory region of the *SLC11A1* gene in yaks.

Haplotype	−1264 A>C	−1237 A>G	−1057 G>T	−127 G>A	−96 A>G	−50 G>A	Number	Frequency
1	A	A	G	G	A	G	152	0.854
2	C	A	G	G	G	G	8	0.045
3	A	A	G	A	A	G	2	0.011
4	C	A	G	A	A	G	2	0.011
5	A	A	G	A	G	A	1	0.005
6	A	A	G	A	G	G	1	0.005
7	A	G	T	A	G	A	1	0.005
8	A	G	T	A	G	G	1	0.005
9	A	G	T	G	G	A	1	0.005
10	A	G	T	G	G	G	1	0.005
11	C	A	G	A	G	A	1	0.005
12	C	A	G	A	G	G	1	0.005
13	C	G	T	A	G	A	1	0.005
14	C	G	T	A	G	G	1	0.005
15	C	G	T	G	G	A	1	0.005
16	C	G	T	G	G	G	1	0.005
Total							176	0.981

**Table 4 animals-13-03615-t004:** Linkage disequilibrium analysis of SNPs in the 5′ regulatory region of the *SLC11A1* gene in yaks.

D′	−1237 A>G	−1057 G>T	−127 G>A	−96 A>G	−50 G>A
−1264 A>C	0.447	0.447	0.596	1.000	1.000
−1237 A>G	-	1.000	0.457	1.000	0.551
−1057 G>T	-	-	0.457	1.000	0.551
−127 G>A	-	-	-	0.674	1.000
−96 A>G	-	-	-	-	1.000

*D’* > 0.75 indicates a tight linkage of SNPs’ loci; *D’* < 0.75 indicates a weak linkage of SNPs’ loci.

## Data Availability

Data are contained within the article.

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
