# Peer review of "Molecular Cloning and Functional Characterization of the 5′ Regulatory Region of the SLC11A1 Gene from Yaks"

_animals, 2023, doi:10.3390/ani13233615_

Round 1

Reviewer 1 Report

Comments and Suggestions for Authors

Molecular cloning and functional characterization of the 5' reg-2 ulatory region of the SLC11A1 gene from Yak

The manuscript is well written but I have the following comments:

Major

1.       Methods: Was DNA quality assessed?

2.       Table 2. The meaning of PIC is not clear to me. Also the reference to ‘High polymorphism’ does not make sense – do the authors mean highly polymorphic?

3.       Fig 6 – it is not clear which vector contains which allele, can this be made clear?

Minor

1.       L14 What does ‘renowned’ mean here?

2.       L18: To ‘identify TFBS, not anticipate

3.       L19 ‘Entrusted’ is not an appropriate term here

4.       L21 region, not territory. Not ‘fashion ’but ‘design’

5.       L29 needs rewording – it seems that some software has been used to offer alternative English words but they do not make sense in the context in which they are used – such as ‘profound exploration’?? (many other examples including L49, L66 (vital life processes)). The authors need to use more appropriate scientific terminology.

6.       L31 What is ‘instantaneous transfection’?

7.       L99 What sequencing results – there is no reference to where the DNA was sequenced?

8.       L111 Is rumen cattle an actual breed? This text is repeated in L121

9.       L191 What is a ‘rigorous’ ANOVA?

10.   L230 ‘Was conducted’, is this Genbank? This section sounds like work they are planning to do, rather than already done. The authors need to check this carefully.

Comments on the Quality of English Language

As above

Reviewer 2 Report

Comments and Suggestions for Authors

The authors mostly rely on the bioinformatics supported by luciferase reporter assay. However, such types of promoter region studies need authentication through EMSA and ChiP assay to validate DNA protein interaction and fully explore roles of the Transcription factor binding sites in the regulation of the target gene. Hence I will suggest to add EMSA, ChiP assay and siRNAs of the selected TFs in the regulation of the target gene 

Comments on the Quality of English Language

Needs editing for English language 

Reviewer 3 Report

Comments and Suggestions for Authors

Lines 50-51: what about the knowledge already available for the yak SLC11A1 gene (not the regulatory region but just the portion of the genome where the pre-mRNA is located)?

Line 70: why did you collect exactly 89 spleen samples?

Lines 78-82: the sequence of this part of the yak genome is available? If yes please write the corresponding ID number(s). And, if it is already available, why do you perform the cloning and analysis of the same region instead of identifying directly in the reference sequence the typical motifs included in the promoter region (paragraph 3.4)? Finally, do you amplify all 89 samples?

Lines 110-111 and 120-122: please include the ID numbers of the sequences you retrieved from the NCBI database here or in a table.

Lines 111-112: rumen cattle?

Line 135: “the company” which Company?

Line 143: this is the second time you used RepeatMask. Please indicate the differences in its use here and as reported in line 128.

Line 172: DMEM?

Line 173: LPS?

Line 173: IFN?

Line 185: LAR?

Lines 205-207: this is materials and methods. The same consideration is valid in other parts of the results section.

Figure 1: why do you use these molecular weight markers? They cannot indicate with high precision the dimension of the amplified fragments.

Lines 321-322: the evidence you wrote (frequency of 0.854) is expected because this haplotype is constituted of the most frequent alleles at each SNP you found.

Table 4: can you explain why you found that some SNPs are in weak linkage?

Lines 334-335: this conclusion is completely NOT supported by any experimental evidence.

Discussion: before discussing the findings you obtained you must rewrite the results section. This is valid for a completely new version of the manuscript.

Comments on the Quality of English Language

Please check carefully the English language, the verbs, and the words you used.

Round 2

Reviewer 2 Report

Comments and Suggestions for Authors

Instead of resubmitting the same research, the effects should be made to include more research such as EMSA, ChIP assay, and silencing or over expression of the TF binding sites and then its role in the regulation of the SLC11A1 gene.

Comments on the Quality of English Language

Minor correction 

Reviewer 3 Report

Comments and Suggestions for Authors

The manuscript is now improved but before it is ready for publication several questions (reported below) must be answered by the authors.

Line 11: remove the full stop after “population”.

Lin3 37: expanse?

Line 63: intron 9 (or exon 9)

Line 68: why “random”?

Line 70: “relatively good haplotype”; good?

Line 77: Bagheri et al. > Bagheri et al. (2016)

Line 81: “mastitis related disease resistance gene” please check

Lines 81-82: “Professor Zhang from Northwest A&F University” > Gao et al. (2017) ???

Lines 86-87: the lacuna exists also for the coding region of this gene in Yak.

Answer to Question 3: ENSBTAG00000015520 I think it is a Bos taurus sequence, not a Yak one! ENSBMUG00000025898 is not found in Google. I think you confused the coding region with the 5’ regulatory region (I think you consider the prompter region with this name not the 5’ regulatory region included in the mRNA). Please complete, if possible, lines 116-118. The SNPs detection is described in paragraph 2.5

Line 149: yellow cattle? (Bos taurus?)

Paragraph 2.4: All these analyses could be performed using a reference sequence.

Line 171: Sanger sequencing?

Paragraph 2.5 have you checked the online tool VEP?

Answer to question 5: Zebu is not a cattle breed!

Answer to question 7: I have not found the correction you wrote.

Lines 331-332: “low mutation rate” or nearly fixed alleles?

Several times: “P” or “p” values? Italic or not?

Line 352 “half” (two times)? Please write exactly the SNPs you refer to here or in the table.

Answer to question 15: in table 4 are reported some SNPs you found in a single locus, not in many different genes! The differences in linkage disequilibrium you found must be better explained. Have you checked the linkage disequilibrium among the “remaining half” SNPs?

Line 388: Zhang Libo ???

Lines 413-414: “This suggests the potential existence of fundamental promoter functions within this sequence” already described for the same region in cattle.

Line 516: “this experiment opted” the options are chosen by the researchers, not by the experiment!

Please avoid repeating parts of the results in the discussion section in order to shorten the text. Some parts I think can be more useful in the introduction.

Line 523: 1418 or -1418?

Lines 603-604: “possibly due to sample size considerations” ???

The putative function of the SNPs you found in the 5’ regulatory region must be better described and supported by strong evidence.

Comments on the Quality of English Language

Please check the English language and the words you used.
